# Fatigue Crack Growth under Non-Proportional Mixed Mode Loading in Rail and Wheel Steel Part 1: Sequential Mode I and Mode II Loading

## Makoto Akama

Department of Mechanical Engineering for Transportation, Osaka Sangyo University, Osaka 574-8530, Japan; akama@tm.osaka-sandai.ac.jp; Tel.: +81-72-875-3001

**Abstract:** Fatigue tests were performed to estimate the coplanar and branch crack growth rates on rail and wheel steel under non-proportional mixed mode I/II loading cycles simulating the load on rolling contact fatigue cracks; sequential and overlapping mode I and II loadings were applied to single cracks in the specimens. Long coplanar cracks were produced under certain loading conditions. The fracture surfaces observed by scanning electron microscopy and the finite element analysis results suggested that the growth was driven mainly by in-plane shear mode (i.e., mode II) loading. Crack branching likely occurred when the degree of overlap between these mode cycles increased, indicating that such degree enhancement leads to a relative increase of the maximum tangential stress range, based on an elasto–plastic stress field along the branch direction, compared to the maximum shear stress. Moreover, the crack growth rate decreased when the material strength increased because this made the crack tip displacements smaller. The branch crack growth rates could not be represented by a single crack growth law since the plastic zone size ahead of the crack tip increased with the shear part of the loading due to the $T$-stress, resulting in higher growth rates.

**Keywords:** non-proportional mixed mode loading; fractography; mode II stress intensity factor; finite element analysis; rail steel; wheel steel

---

## 1. Introduction

When railway wheels repeatedly pass over the rails, rolling contact fatigue (RCF) cracks occur on both the wheel tread and the railhead. The accumulation of large plastic shear strains generates these cracks, which initially grow in the plastic flow direction of the resulting microstructure [1]; when their growth reaches a depth of several millimeters, they are unaffected by the plastic deformation layer and are then subjected to a mixed mode I/II/III loading, growing as coplanar cracks at shallow angles to the surface [2–5]. The mode II/III loading cycle is almost proportional, while the mode I/II and I/III cycles are non-proportional.

Fremy et al. [6] conducted experiments in non-proportional mixed mode I/II/III loading conditions to investigate the load path effect on fatigue crack growth (FCG) in 316L stainless steel. They applied the 'star' load path to a cruciform specimen containing an inclined crack by using a six-actuator servo-hydraulic testing machine; the addition of mode III loading steps to a mode I/II loading sequence increased the FCG rate even when the crack path was not significantly modified. However, the tested material was not rail or wheel steel and the load history did not simulate the one experiencing RCF cracks. Moreover, only a coplanar crack growth below 2 mm was obtained.

Many studies focused on the coplanar crack growth of rail steel under non-proportional mixed mode I/II loading cycles. Figure 1 shows how mode I and II loadings generated in the rail and wheel when the fluid penetrates into the crack. Details will be described in the next section. Bold et al. [7]

performed fatigue tests, sequentially applying cyclic mode I and II loadings on rail steel biaxial cruciform specimens. When they removed the mode I loading before applying the fully reversed mode II and the ratio between the nominal range values of the mode I and II stress intensity factors (SIFs), $K_I$ ($\Delta K_I$), and $K_{II}$ ($\Delta K_{II}$), were greater than or equal to 0.5, a long coplanar crack growth was observed. Wong et al. [8,9] carried out sequential mixed mode I/II loading experiments on cruciform specimens; when they applied mode II loading before removing mode I, a degree of overlap ($\delta$) appeared. The effective values of $\Delta K_I$ and $\Delta K_{II}$ ($\Delta K_{Ieff}$ and $\Delta K_{IIeff}$, respectively) and the $\delta$ effect on the crack growth were considered and crack growth models were proposed in the $\Delta K_{Ieff}$ and $\Delta K_{IIeff}$ forms. They reported that $\Delta K_{Ieff}$ is a control parameter determining the crack growth direction and that the $\delta$ increase encourages crack branching; they also proposed a branch criterion in terms of $\Delta K_{Ieff}$ and $\delta$. Akama and Susuki [10] conducted non-proportional mixed mode I/II FCG tests on rail and wheel steels in practical use in Japan. Based on the study of Wong et al., they defined crack growth models using $\Delta K_{Ieff}$ and $\Delta K_{IIeff}$ and found that the cracks branched easily in wheel steel compared to rail steel. They also examined the fracture surfaces, but no clear striation pattern was observed.

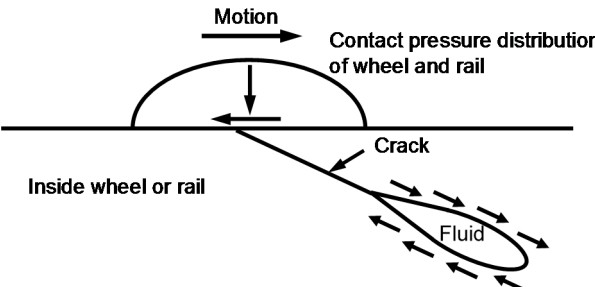

**Figure 1.** Schematic representation of how a fluid in the crack generates mode I and II loadings.

Sequential mode I/II FCG tests on rail steel were also performed by Doquet and Pommier [11]. They used tubular specimens and gold microgrids with a 4 mm pitch, which were laid ahead of the crack tips and along the crack surfaces to observe the residual shifts; hundreds of microns of constant-speed coplanar growth were obtained when $\Delta K_{II}/\Delta K_I$ ranged from 1 to 4. They also investigated the role of plasticity and crack face friction via finite element analysis (FEA) and concluded that the maximum growth rate criterion rationalized the crack path observed in the non-proportional loadings. Doquet et al. [12] investigated through FEA the influence of static and 90° out-of-phase cyclic mode II loading superimposed to the cyclic mode I on the plasticity- and asperity-induced closure for an aluminum alloy. The applied mode II could either increase or decrease $\Delta K_{Ieff}$ depending on the stress ratio and loading path; in some cases, it had opposite effects on the two closure mechanisms. In the case of 90° out-of-phase mixed mode loading, mode II drastically reduced or even suppressed the plasticity-induced closure. To clarify the incipient direction of the crack path observed by Bold et al. [8], Dahlin and Olsson [13] investigated several criteria while using the FEA approach; they concluded that a maximum tangential stress (MTS) range criterion based on an elasto–plastic stress field can appropriately predict the incipient crack growth direction in ductile metals. Yu et al. [14] studied the FCG behavior in a thin-walled tubular specimen of aluminum alloy under non-proportional mixed mode I and II loading, showing that the FCG significantly differed from that under mode I or proportional mixed mode loading. Although they obtained a long and stable shear mode growth, they could not apply the commonly accepted MTS criterion in most of the non-proportional loading cases.

As indicated above, previous studies on coplanar crack growth under non-proportional mixed mode I/II loading cycles apparently focused on rail steel. Only Akama and Susuki [10] considered also the wheel steel case, but their elucidation of the phenomena associated with crack growth under such loading cycles was not sufficient. In particular, the reason why the growth direction of the crack changes depending on the loading conditions and the type of steel was not clarified. Besides, the crack growth models by Wong et al. [9] have been pointed out as arbitrary. Therefore, in this study, crack

growth data from non-proportional mode I/II loading cycles were re-constructed by using a reliable equivalent SIF range and FEA was performed to elucidate the crack growth behavior under these loading cycles. The reliable criteria for predicting the crack path direction of non-proportional mixed mode loading were used, and the results were compared with corresponding experimental results.

This paper, part 1 of two companion papers, presents FCG under non-proportional mixed mode I/II loading and is organized as follows. The introduction starts in this section with a brief overview of past papers for the crack growth under non-proportional mixed mode I/II loading cycles. Section 2 describes the detailed method of the experiments conducted and presents the experimental results. In Section 3, the FEA model for predicting the crack path direction is presented and the results are indicated. Section 4 gives detailed considerations and discussions by comparing the experimental and FEA results. Finally, the important results obtained in this study are summarized in Section 5.

## 2. Experiments

### 2.1. Testing Machine

The crack growth under mixed mode I/II loading was investigated with an in-plane biaxial testing machine consisting of four hydraulic actuators, each one having a 200 kN tension–compression capacity for static and fatigue loads. The actuator assemblies formed two pairs of X and Y axes and were rigidly and diagonally mounted in an octagonal box-shaped frame, which was of heavy-duty welded construction and placed horizontally. A load cell and a linear variable differential transformer were installed in each actuator assembly. The components also included manifolds, a control console, and a hydraulic pump unit. To obtain and maintain the designated stress ratios during the fatigue tests, an improved control system was developed specifically for this study. The two actuator pairs were controlled by both load and positioning feedback signals. The control system of the load signals X1 and X2 for the *x*-axis actuator pair is schematized in Figure 2. The load command signal FX was derived from the load feedback signals FX1 and FX2 and from the position feedback signals S1 and S2, which were calculated, correspondingly, from F1 and F2 and from the calculation circuits CX1 and CX2; the notations F, S, 1, and 2 indicate, respectively, the load cell, the differential transformer output, and the X1 and X2 actuators. The same method was applied for the *y*-axis actuator. Testing frequencies from 1 to 2 Hz were used.

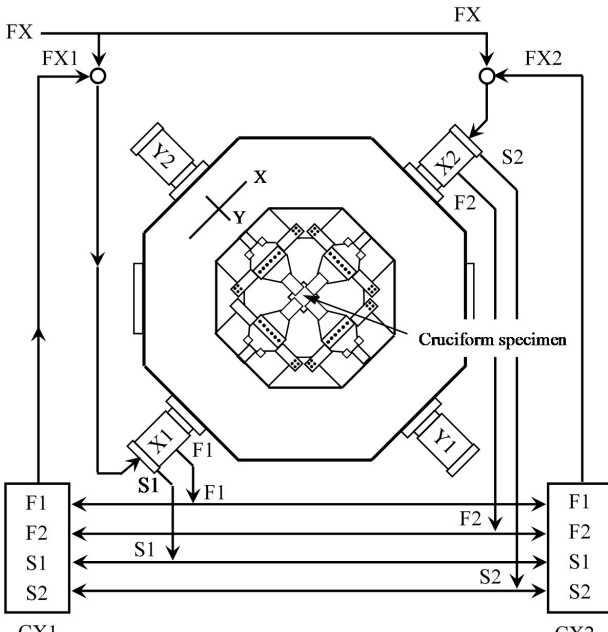

**Figure 2.** Schematic of the in-plane biaxial testing machine and the system controlling the load signals for the actuators.

## 2.2. Specimens

Cruciform specimens (Figure 3) having the 45° starter notch with a half-length of 2 mm, made by spark erosion, were used in the experiments [15]. Each specimen had a uniformly stressed square working section (72 × 72 mm, 4 mm in thickness) to observe crack growth behaviors. The tensile and compressive loads were applied, respectively, through pins and at the specimen edge; the backlash was removed via wedge tightening. The starter notch was pre-cracked by using an equibiaxial mode I loading with a zero stress ratio. The load was reduced during pre-cracking in 10% steps until an FCG rate of less than $10^{-9}$ m/cycle was reached. This was to reduce the residual plastic zone size to less than that produced during the first cycle of the test.

Two-rail steels (RP and RF) and one-wheel steel (WT) were used as specimen materials. The RP and RF were used as normal and head hardened rails, respectively, that were installed for corresponding straight sections and outer rails in curved sections. Their chemical compositions and mechanical properties are summarized in Tables 1 and 2, respectively, showing that the ultimate tensile strength and 0.2% proof stress of RF were much superior to those of RP and WT. The microstructures of RP and WT were normal pearlite, whereas that of RF was fine pearlite with an average lamellar spacing of about 100 nm. The specimens were collected directly from real rails and wheels.

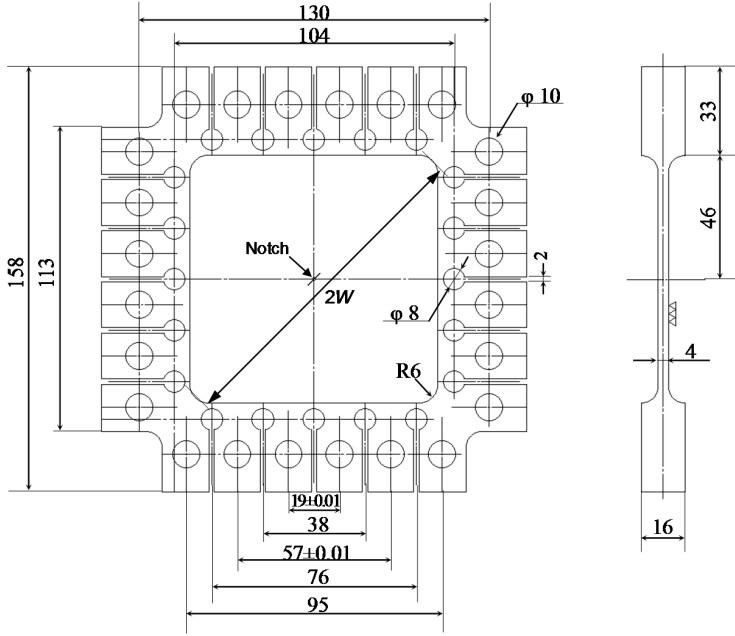

(**a**) Whole configuration (all dimensions in mm)

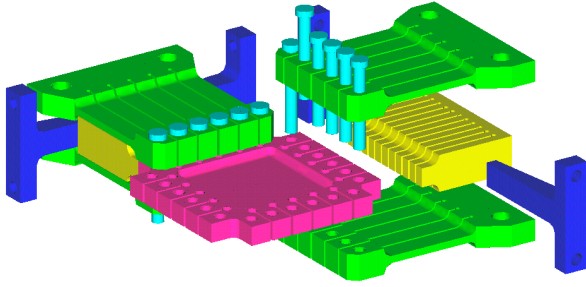

(**b**) Arrangement of specimen, tensile and compressive grips, pin and wedges

**Figure 3.** (**a**) Configuration of the cruciform specimen used in the experiments (all dimensions in mm) and (**b**) its arrangement with the tensile and compressive grips, pin, and wedges.

**Table 1.** Chemical composition of steels used for the experiments (wt%).

| Material | C | Si | Mn | P | S |
|---|---|---|---|---|---|
| Rail steel, RP | 0.68 | 0.26 | 0.93 | 0.016 | 0.01 |
| Rail steel, RF | 0.79 | 0.17 | 0.82 | 0.019 | 0.01 |
| Wheel steel, WT | 0.65 | 0.26 | 0.73 | 0.016 | 0.01 |

**Table 2.** Mechanical properties.

| Material | Ultimate Tensile Strength (MPa) | 0.2% Proof Stress (MPa) |
|---|---|---|
| Rail steel, RP | 934 | 511 |
| Rail steel, RF | 1214 | 802 |
| Wheel steel, WT | from 981 to 1030 | from 618 to 657 |

### 2.3. Loading History

The loading history, illustrated in Figure 4, simulated that experienced by RCF cracks in the presence of a fluid, as obtained by FEA [2–5]. Since the load is represented by sine waves, the time on the horizontal axis is indicated as an angle (degree); the figure also shows $\delta$ between $K_I$ and $K_{II}$. When a contact area between the wheel and the rail approaches a surface crack, the surface tangential traction slightly opens the crack mouth in the driven surface and the surrounding fluid penetrates inside it. When the contact area moves over the crack mouth, the $K_I$ load increases (0–180°) due to the fluid entrapment [16] or the hydraulic pressure mechanism [17]; after it leaves the crack mouth, the fluid in it flows out and its influence on $K_I$ almost disappears (180°). Meanwhile, the $K_{II}$ loading is also generated and completely reversed (90–450°). Hence, if there is fluid inside the crack, cracks in the triaxial compressive stress field can open in the contact area and large $K_I$ and $K_{II}$ loading are generated.

This condition was simulated by actuating both axes of the biaxial machine to apply the required mode I and II loading sequence to the crack. Figure 5 shows loading examples of the *x*- and *y*-axes for a 45° crack.

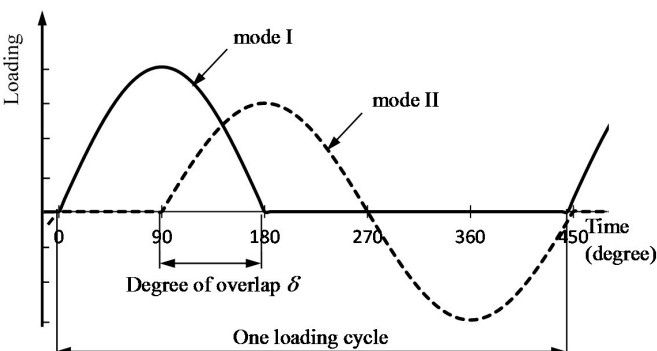

**Figure 4.** Loading cycle applied under the condition of $\Delta K_{II}/\Delta K_I = 1.5$ and $\delta = 90°$.

### 2.4. Calculation of the Stress Intensity Factors

$K_I$ and $K_{II}$ were calculated as

$$K_I = Y\sigma\sqrt{\pi a} \tag{1}$$

$$K_{II} = Y\tau\sqrt{\pi a} \tag{2}$$

where $Y = \sqrt{\sec(\pi a/2W)}$ for the central 45° crack in a plate of diagonal length 2W (refer to Figure 3) and $\sigma$ and $\tau$ are the normal and in-plane shear stress, respectively, applied to the crack. The crack length (*a*) was measured with a traveling microscope at the resolution of 1 μm. In addition, $\Delta K_{Ieff}$ and $\Delta K_{IIeff}$ were defined by the ratios of crack closure ($U_I$) and locking ($U_{II}$), respectively [18]

$$\Delta K_{\text{Ieff}} = U_{\text{I}} \Delta K_{\text{I}} \tag{3}$$

$$\Delta K_{\text{IIeff}} = U_{\text{II}} \Delta K_{\text{II}} \tag{4}$$

$$U_{\text{I}} = \frac{v_{a\max} - v_{a\min}}{v_{th\max} - v_{th\min}} \tag{5}$$

$$U_{\text{II}} = \frac{u_{a\max} - u_{a\min}}{u_{th\max} - u_{th\min}} \tag{6}$$

where $v_{a\max}$, $v_{a\min}$, $v_{th\max}$, and $v_{th\min}$ are the measured maximum, measured minimum, theoretical maximum, and theoretical minimum opening displacements, respectively, and $u_{a\max}$, $u_{a\min}$, $u_{th\max}$, and $u_{th\min}$ indicate the corresponding values for the sliding displacement. The crack closure and locking were measured via the surface replica technique.

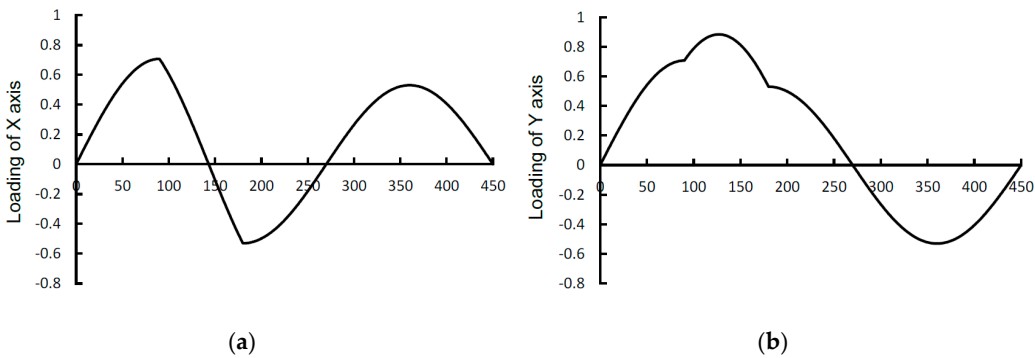

(**a**)　　　　　　　　　　　　　　　　　　　(**b**)

**Figure 5.** Sequential loading cycle of the (**a**) *x* and (**b**) *y* axes under the condition of $\Delta K_{\text{II}}/\Delta K_{\text{I}} = 1.5$ and $\delta = 90°$; the vertical axis represents the relative values of the loads of *x*- and *y*-axes.

## 2.5. Experimental Conditions

The experiments were performed on four WT, six RP, and one RF specimen and, hereafter, will be accordingly referred to as WTl, WT2, WT3, and WT4; RP1, RP2, RP3, RP4, RP5, and RP6; and RF1; respectively. The stress ratios of mode I and mode II loading ($R_{\text{I}}$ and $R_{\text{II}}$) were 0 and −1, respectively. Table 3 shows the $\Delta K_{\text{II}}/\Delta K_{\text{I}}$ ratios and $\delta$ values for each experiment; WT1–WT4; RP1–RP4, and RF1 were designed to mainly produce coplanar crack growth rate data, whereas RP5 and RP6 were conducted for obtaining branch crack growth rate data.

**Table 3.** Testing conditions.

| Exp. No. | $\Delta K_{\text{II}}/\Delta K_{\text{I}}$ | $\delta$(degree) |
|---|---|---|
| WT1 | 1.0 | 0, 10, 20, 30, 40, 50, 60, 70, 80, 90, 120 |
| WT2 | 1.0 | 30, 60, 90, 120 |
| WT3 | 1.25 | 0, 10, 20, 30, 40, 50, 60, 70, 80, 90, 120 |
| WT4 | 1.0, 1.375, 1.5, 1.9, 2.0 | 10, 30, 60 |
| RP1 | 1.0 | 0, 10, 20, 30, 40, 50, 60, 70, 80, 90, 120 |
| RP2 | 1.5 | 0, 10, 20, 30, 40, 50, 60, 70, 80, 90 |
| RP3 | 1.0 | 30, 60, 90, 120, 150 |
| RP4 | 1.5 | 30, 60, 90, 120 |
| RP5 | 2.0 | 0 |
| RP6 | 2.5 | 0 |
| RF1 | 1.0, 1.375, 1.5 | 30 |

*2.6. Experimental Results*

A polar coordinate system having the origin coinciding with the crack tip was introduced, as shown in Figure 6 together with a schematic representation of the main coplanar crack plane and its angle ($\theta$) with the branch crack plane, when branching occurs.

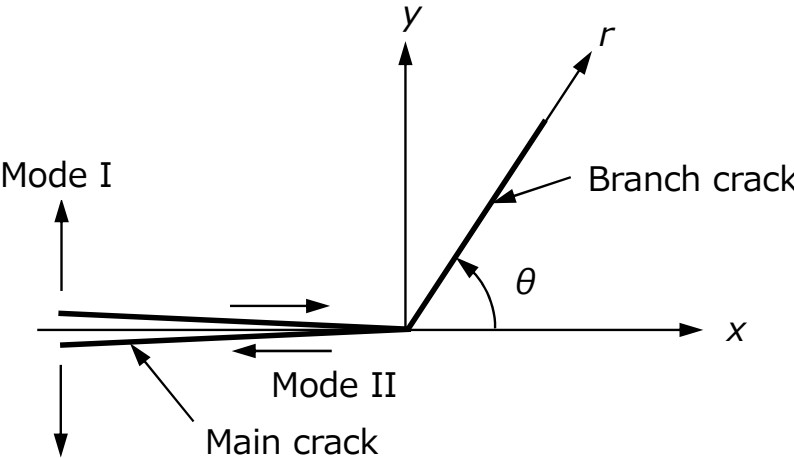

**Figure 6.** Branch crack at a main crack plane and branch angle definition.

The coplanar crack grew almost straight (i.e., $\theta = 0$) for all $\delta$ values in WTl–WT3, RP1–RP3, and RF1. After the crack length on each side extended to about 3 mm, $\delta$ increased during 10° or 30° steps. The $\Delta K_I$ and $\Delta K_{II}$ values were maintained approximately constant for each loading step by decreasing the loads. In WT4, the coplanar crack grew when $\Delta K_{II}/\Delta K_I = 1.0$ regardless of $\delta$, but it branched when this ratio exceeded 1.375. Immediately after this branch crack growth, the crack path was restored to an initial 45° plane by using an equibiaxial mode I loading and, then, the experiments were continued. In RP4, the coplanar crack branched at a $\theta$ of about –70° when $\delta$ was 120°. When the crack was short, $U_I$ and $U_{II}$ at the crack tip were hard to estimate. However, some $v_{amax}$, $v_{amin}$, $u_{amax}$, and $u_{amin}$ measurements were performed near the crack tip and the ratios $U_I$ and $U_{II}$ were estimated by extrapolating back to the crack tip. In RP5 and RP6, branch cracks grew; for these branched cracks, the displacements to obtain $U_I$ and $U_{II}$ were not measured.

2.6.1. Coplanar Crack Growth Rate

The coplanar crack growth rates of RP, which comprises RP1, RP2, and RP3 obtained from the mixed mode I/II tests, were plotted against $\Delta K_I$ (Figure 7), including all $\delta$ cases and, for comparison, the pure mode I growth data collected at $R_I = 0$. The FCG rates for the mixed mode I/II loading were fairly faster than those for pure mode I. When the Paris-type law was applied to the crack growth rates for all the mixed mode I/II loading data, we obtained the equation

$$\frac{da}{dN} = 3.75 \times 10^{-12} (\Delta K_I)^{3.39} \tag{7}$$

where $N$ is the number of cycles. The coefficient of determination ($R^2$) was 0.69.

Next, the same growth rates were plotted against $\Delta K_{II}$ (Figure 8), revealing an improvement in $R^2$ (0.87)

$$\frac{da}{dN} = 4.80 \times 10^{-12} (\Delta K_{II})^{3.12} \tag{8}$$

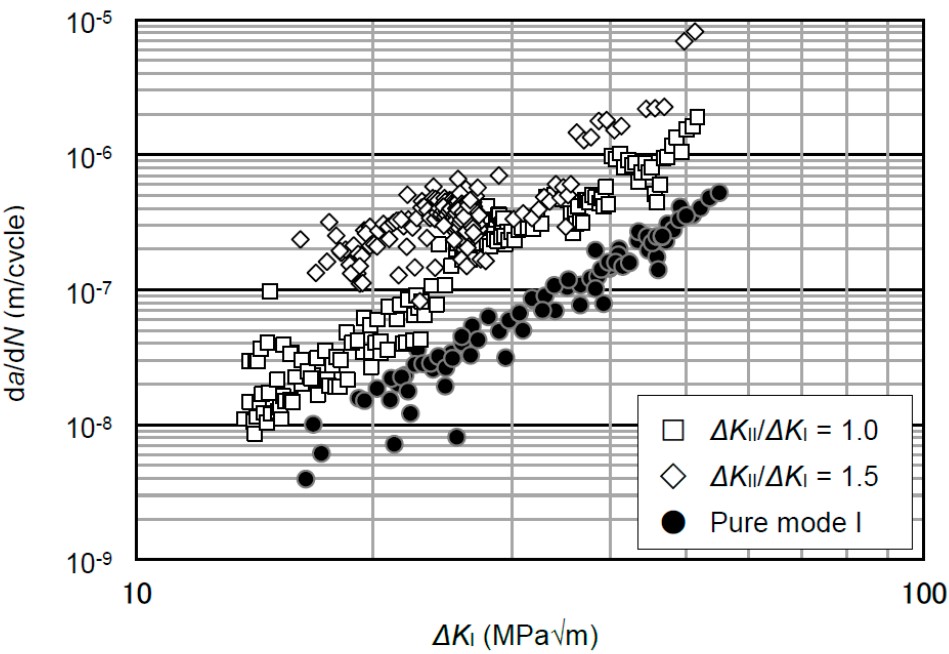

**Figure 7.** Coplanar crack growth rates against $\Delta K_\mathrm{I}$ comparing pure mode I data for RP.

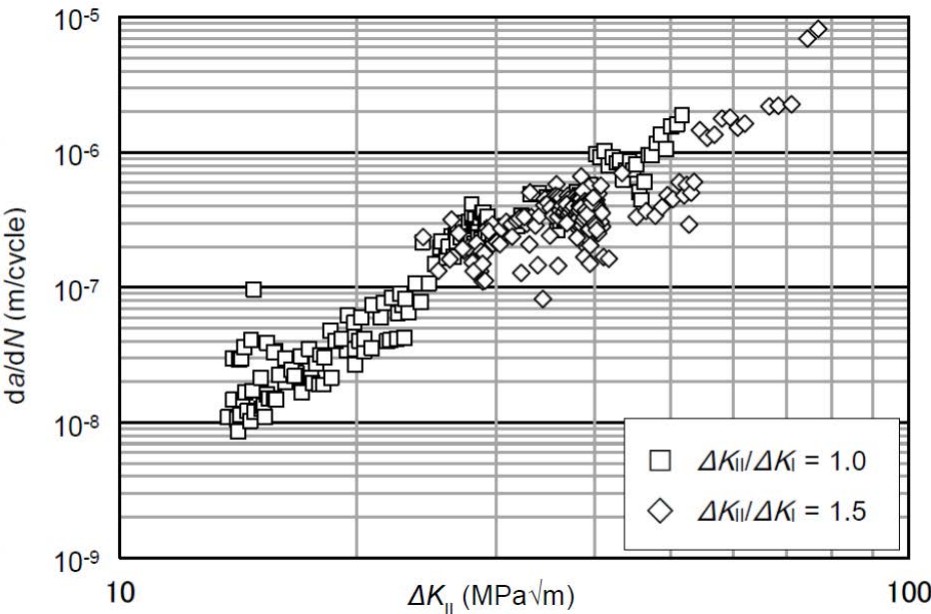

**Figure 8.** Coplanar crack growth rates against $\Delta K_\mathrm{II}$ for RP.

Then, since the crack growth law can be described uniquely even under different loads and stress ratios by effective SIFs, the growth rates were plotted against $\Delta K_\mathrm{Ieff}$ (Figure 9). The crack growth rate data can be summarized by the equation

$$\frac{da}{dN} = 8.19 \times 10^{-11} (\Delta K_\mathrm{Ieff})^{2.59} \tag{9}$$

In this case, a further correlation improvement was obtained ($R^2 = 0.89$). The $\Delta K_{\text{Ieff}}$ and $\Delta K_{\text{IIeff}}$ values were successively used to fit the growth data.

The crack growth rates were then plotted against the equivalent SIF range ($\Delta K_{\text{v}}$), as proposed by Richard et al. [19] (Figure 10), which is defined as

$$\Delta K_{\text{v}} = \frac{\Delta K_{\text{I}}}{2} + \frac{1}{2}\sqrt{\Delta K_{\text{I}}^2 + 4(1.115\Delta K_{\text{II}})^2} \tag{10}$$

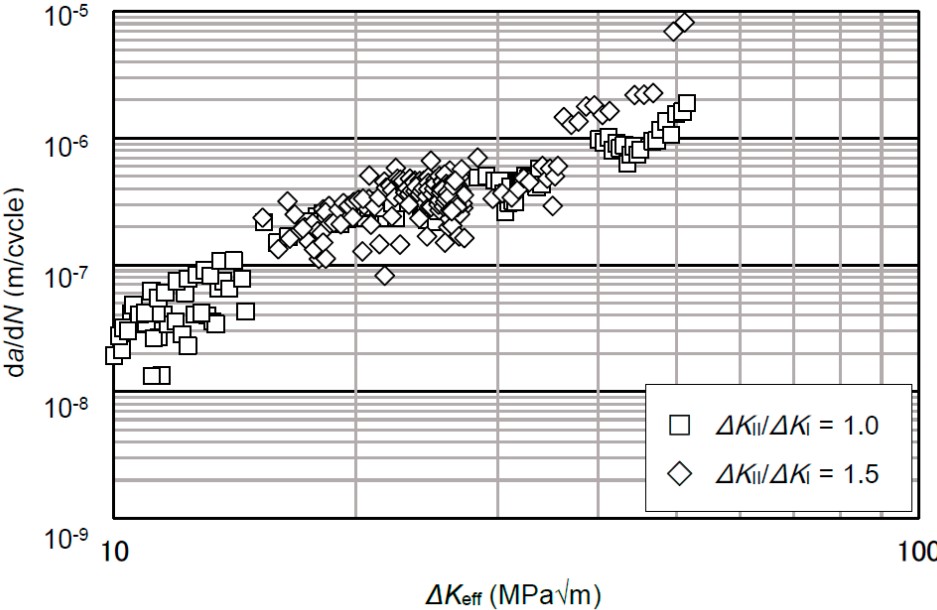

**Figure 9.** Coplanar crack growth rates against $\Delta K_{\text{Ieff}}$ for RP.

The crack growth rates were expressed as

$$\frac{da}{dN} = C(\Delta K_{\text{v}})^m \tag{11}$$

In the RP, RF, and WT cases, the corresponding $C$ and $m$ values were $6.08 \times 10^{-11}$ and 2.18, $8.70 \times 10^{-11}$ and 2.04, and $3.82 \times 10^{-11}$ and 2.45, respectively. Figure 10 shows that better correlations were attained when plotting the crack growth rates against $\Delta K_{\text{v}}$. Over all, WT exhibited the fastest crack growth rates and, among the rail steels, the rates of RP were higher than those of RF.

2.6.2. Branch Crack Growth Rate

In RP, branch cracks occurred and grew where $\Delta K_{\text{II}}/\Delta K_{\text{I}}$ exceeded 2.0 and $\delta$ was 0°. For this specimen geometry, the branch crack was subjected to a large mode I cycle, which was generated from the mode II loading, followed by a small mode I cycle from the mode I loading. Therefore, the growth rates were plotted against the $\Delta K_{\text{I}}$ values from the mode II loading cycles only; $\Delta K_{\text{I}}$ was calculated using the procedure proposed by Gao et al. [20], which considers the stresses acting normal to the crack plane. In particular, the branch crack growth rates of RP were plotted against $\Delta K_{\text{I}}$ (Figure 11). These rates were expressed as

$$\frac{da}{dN} = 1.11 \times 10^{-11}(\Delta K_{\text{I}})^{2.40} \tag{12}$$

for RP5 and as

$$\frac{da}{dN} = 2.64 \times 10^{-10}(\Delta K_{\mathrm{I}})^{1.95} \tag{13}$$

for RP6. As can be seen, the crack growth rate differs considerably depending on $\Delta K_{\mathrm{II}}/\Delta K_{\mathrm{I}}$, and increasing in the $\Delta K_{\mathrm{II}}/\Delta K_{\mathrm{I}}$ increases the crack growth rate.

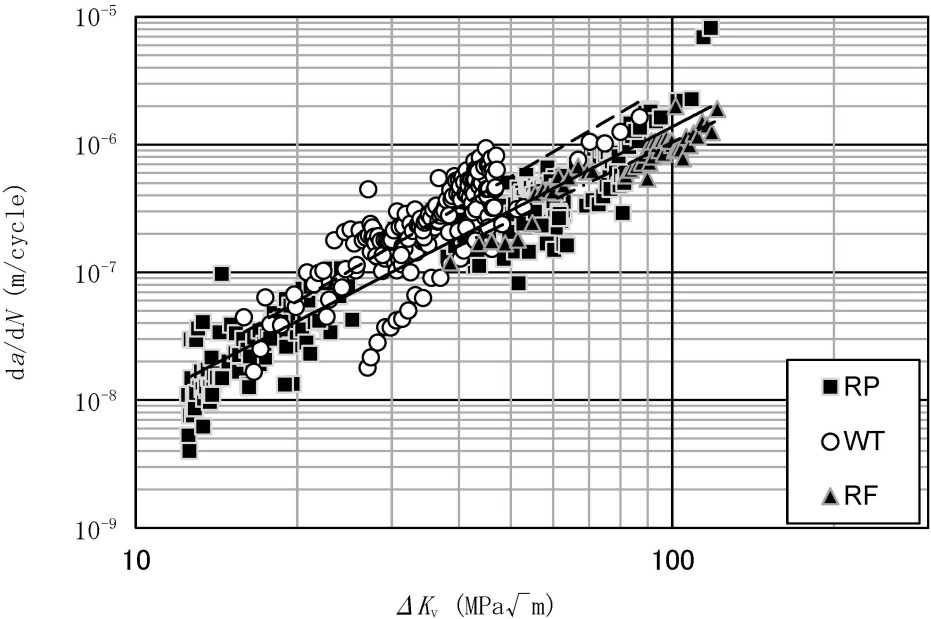

**Figure 10.** Coplanar crack growth rates of mixed mode I/II against $\Delta K_{\mathrm{v}}$ for RP, RF and WT (solid line; RP, one dot chain line; RF, broken line; WT).

### 2.6.3. Fractography

To understand the fracture mechanism, scanning electric microscope (SEM) observations were performed. Figure 12 shows the fracture surfaces near the crack tip region and along the crack flank resulting from RP2, WT3, and RF1. Heavily deformed ridges and valleys were rubbed out in the sliding direction and also several oxidized wear particles were found on the surface. No striation pattern was observed near the crack tip region in case of WT3 and RF1, but RP2 exhibited obscure striations that seemed to be scattered. Hence, the fractographic observation permitted the determination of the crack growth caused by non-proportional mixed mode I/II loading, which was clearly distinguishable from that generated by pure mode I loading.

In summary, these were the crack growth characteristics observed in rail and wheel steel specimens subjected to non-proportional mixed mode I/II loading:

(1)  When $\Delta K_{\mathrm{II}}/\Delta K_{\mathrm{I}}$ increased, the crack tended to branch;
(2)  As $\delta$ increased, the crack easily branched;
(3)  The coplanar crack growth rates in RF were lower than those in WT and RP;
(4)  The branch crack growth rate varied considerably depending on $\Delta K_{\mathrm{II}}/\Delta K_{\mathrm{I}}$ even in the same material, unlike the coplanar growth rate, could not be correlated by a single line;
(5)  No clear striation patterns were found near the crack tip region.

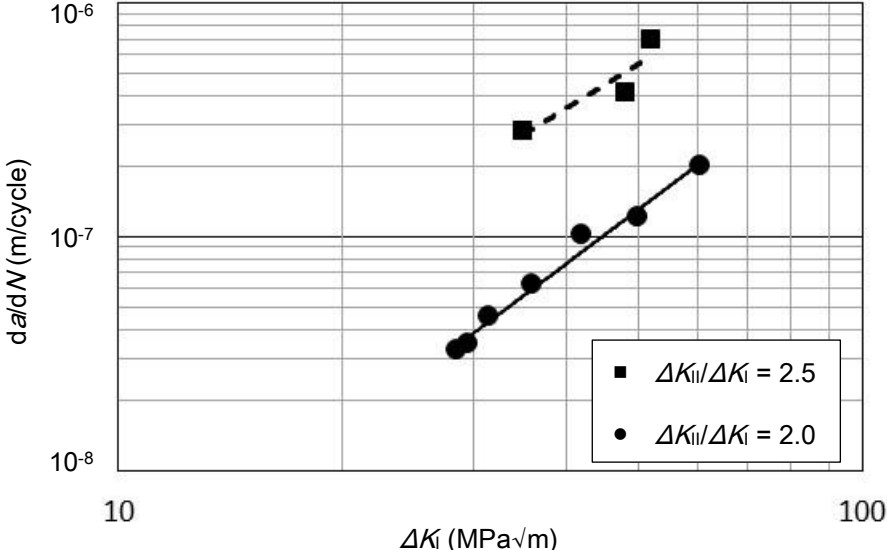

**Figure 11.** Branch crack growth rates of mixed mode I/II against $\Delta K_I$ for RP.

## 3. Finite Element Analysis

### 3.1. Procedure

The elasto–plastic analysis was performed using the commercial FEA code MARC to better elucidate the crack growth characteristics. A stationary crack was considered because, in this study, a branch angle $\theta$ from the main crack was predicted. Figure 13 shows the mesh used, with a crack length $a$ fixed at 5 mm; only the uniformly stressed square working section was considered, represented by a simple square in plain strain assumption with a thickness of 4 mm. Second-order isoparametric quadrilateral and triangular elements, refined to 25 μm near the crack tip, were used with the singular elements [21] arranged around the tips; for smaller elements, they were greatly distorted due to a large number of loading cycles, resulting in remarkably poor analysis accuracy. The total numbers of elements and nodes were 644 and 1968, respectively. The boundary conditions applied for cruciform specimens under mixed mode I/II loading were also described. At each edge center, a pilot node was connected to the remaining nodes of that edge through a multipoint constraint; all the loads were applied to these pilot nodes as concentrated normal and shear forces (*F* and *S*, respectively). The use of this mesh and loading procedure allowed the confirmation that the MTS occurred at $\theta = -70.5°$ due to pure mode II loading in the elastic analysis. The contact between crack faces was taken into account and the friction coefficient was set to 0.3.

When analyzing the fatigue problems, the choice of the material constitutive model is crucial. In this study, the model combining the nonlinear kinematic hardening rule with the isotropic hardening rule, developed by Chaboche and Lemaitre [22] (C&L model), was used as

$$^{ti+\Delta ti}\sigma_y = {}^{0}\sigma_y + Q\left\{1 - \exp\left(-B^{ti+\Delta ti}\bar{e}^p\right)\right\} \tag{14}$$

$$\mathrm{d}\alpha = \frac{2}{3}h\mathrm{d}e^p - \zeta\alpha\mathrm{d}\bar{e}^p \tag{15}$$

where $^{ti+\Delta ti}\sigma_y$ is the updated yield stress at time $t_i + \Delta t_i$, $^{0}\sigma_y$ is the initial yield stress, $Q$, $B$, $h$, and $\zeta$ are material constants, $^{ti+\Delta ti}\bar{e}^p$ is the accumulated effective plastic strain at $t_i + \Delta t_i$, $\alpha$ is the shift of the yield surface center, $e^p$ is the plastic strain, and d of d$\alpha$ and d$\bar{e}^p$ implies increment. The material constants for RP and RF, determined via strain controlled uniaxial fatigue experiments, are summarized in Table 4 along with Young's modulus $E$ and Poisson ratio $\nu$. The FEA was performed on these two rail steels to clarify the material effect on the crack growth rate.

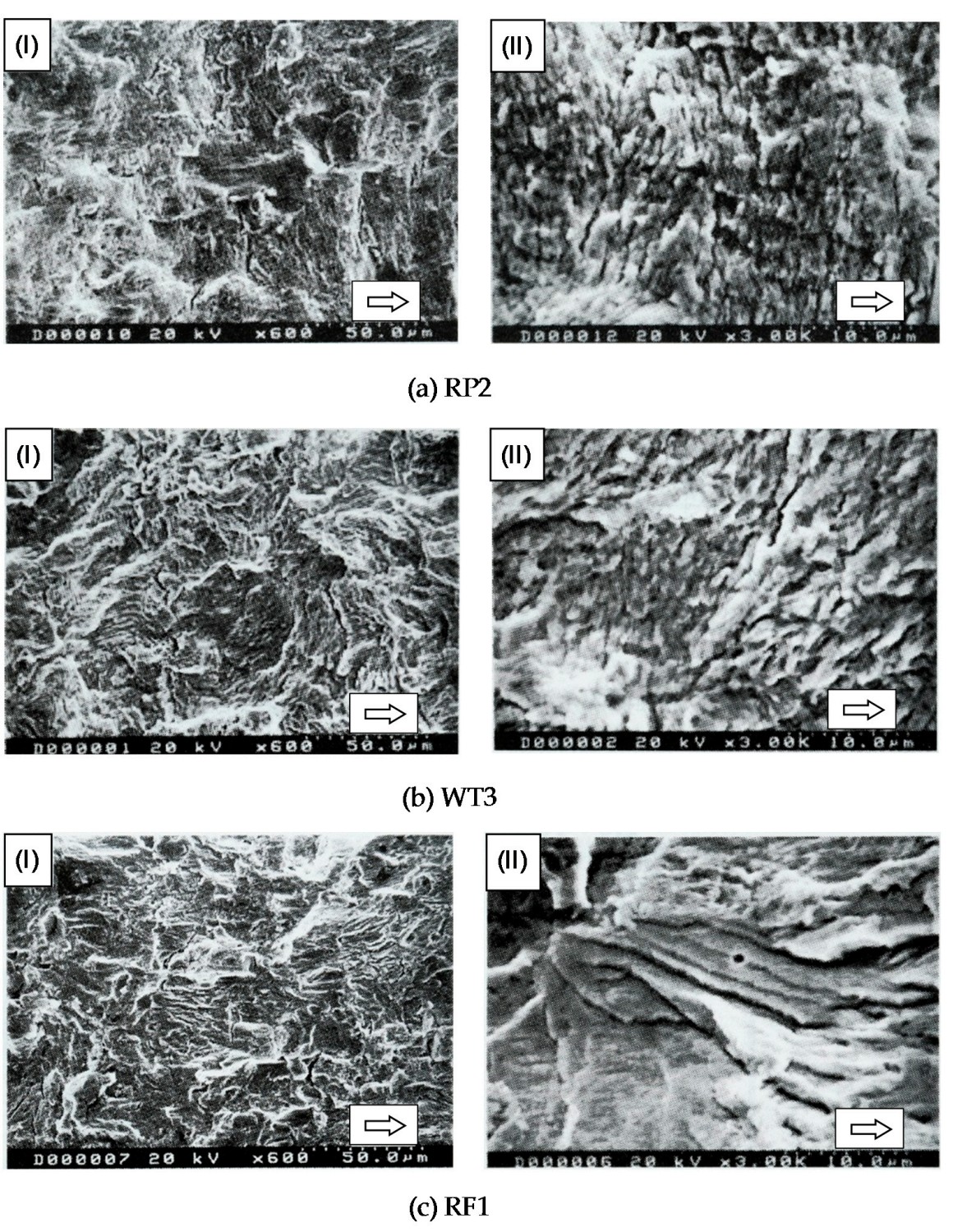

(a) RP2

(b) WT3

(c) RF1

**Figure 12.** Fracture surface near the crack tip; (**a**) RP2, (**b**) WT3, and (**c**) RF1. Arrows indicate the direction of crack growth.

### 3.2. Analytical Results

The analytical conditions (Table 5) were decided based on the actual experiments described in Section 2. Figures 14 and 15 show some results about the distribution of the tangential ($\sigma_\theta$) and shear stresses ($\tau_{r\theta}$), respectively, in the first loading cycle for ARP2; the maximum $\Delta\sigma_\theta$ appeared between $-45°$ and $-67.5°$, whereas the maximum $\Delta\tau_{r\theta}$ was observed exactly at $0°$.

When using the C&L model, several loading cycles are required to achieve a steady-state stress cycle; therefore, at least 100 loading cycles were initially planned for the simulation. However, as the number of cycles increased, the deformation of the crack tip elements increased accordingly and, hence, an accurate analysis was considered no longer possible after the 50th cycle. Thus, the evaluations were performed at the 50th loading cycle and, at this point, a steady-state stress cycle could not be achieved.

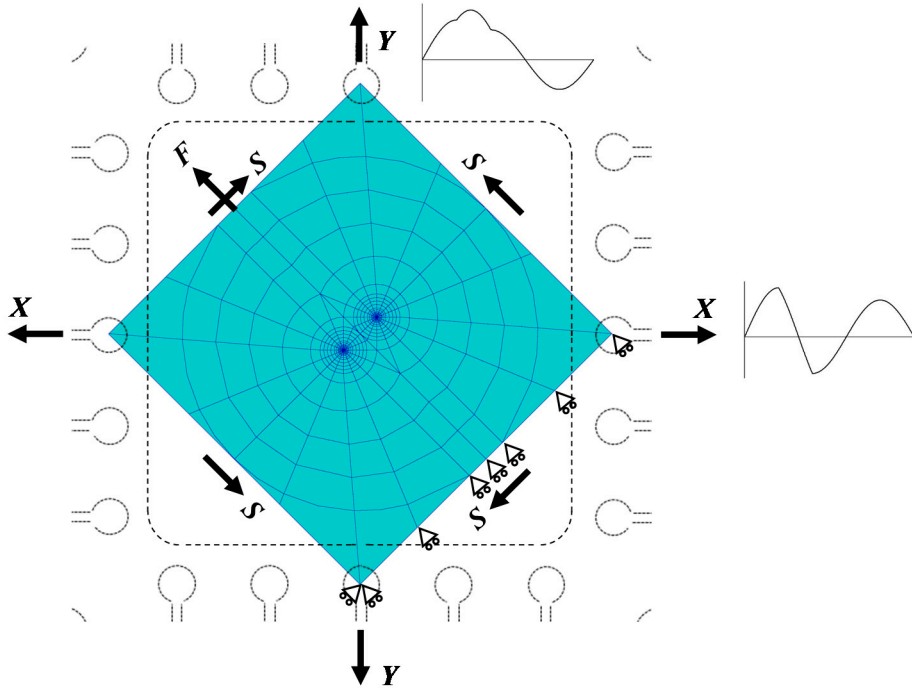

**Figure 13.** Finite element mesh and boundary conditions for the cruciform specimen under mixed mode I/II loading.

**Table 4.** Material properties used in FEA.

|  | $E$ (MPa) | $\nu$ | $^0\sigma_y$ (MPa) | $Q$ | $b$ | $h$ (MPa) | $\zeta$ |
|---|---|---|---|---|---|---|---|
| RP | 183,008 | 0.3 | 508 | −208 | 24.2 | 85,248 | 193 |
| RF | 182,778 | 0.3 | 684 | −264 | 1.27 | 88,615 | 185 |

**Table 5.** FEA conditions.

| No. | $\Delta K_{II}/\Delta K_I$ | $\delta$ (degree) | $F$ (N) | $S$ (N) | Material |
|---|---|---|---|---|---|
| ARP1 | 1.5 | 90 | $0 \leftrightarrow 26{,}667$ | $-20{,}000 \leftrightarrow 20{,}000$ | RP |
| ARP2 | 1.5 | 120 | $0 \leftrightarrow 26{,}667$ | $-20{,}000 \leftrightarrow 20{,}000$ | RP |
| ARP3 | 1.5 | 30 | $0 \leftrightarrow 26{,}667$ | $-20{,}000 \leftrightarrow 20{,}000$ | RP |
| ARF1 | 1.5 | 30 | $0 \leftrightarrow 26{,}667$ | $-20{,}000 \leftrightarrow 20{,}000$ | RF |

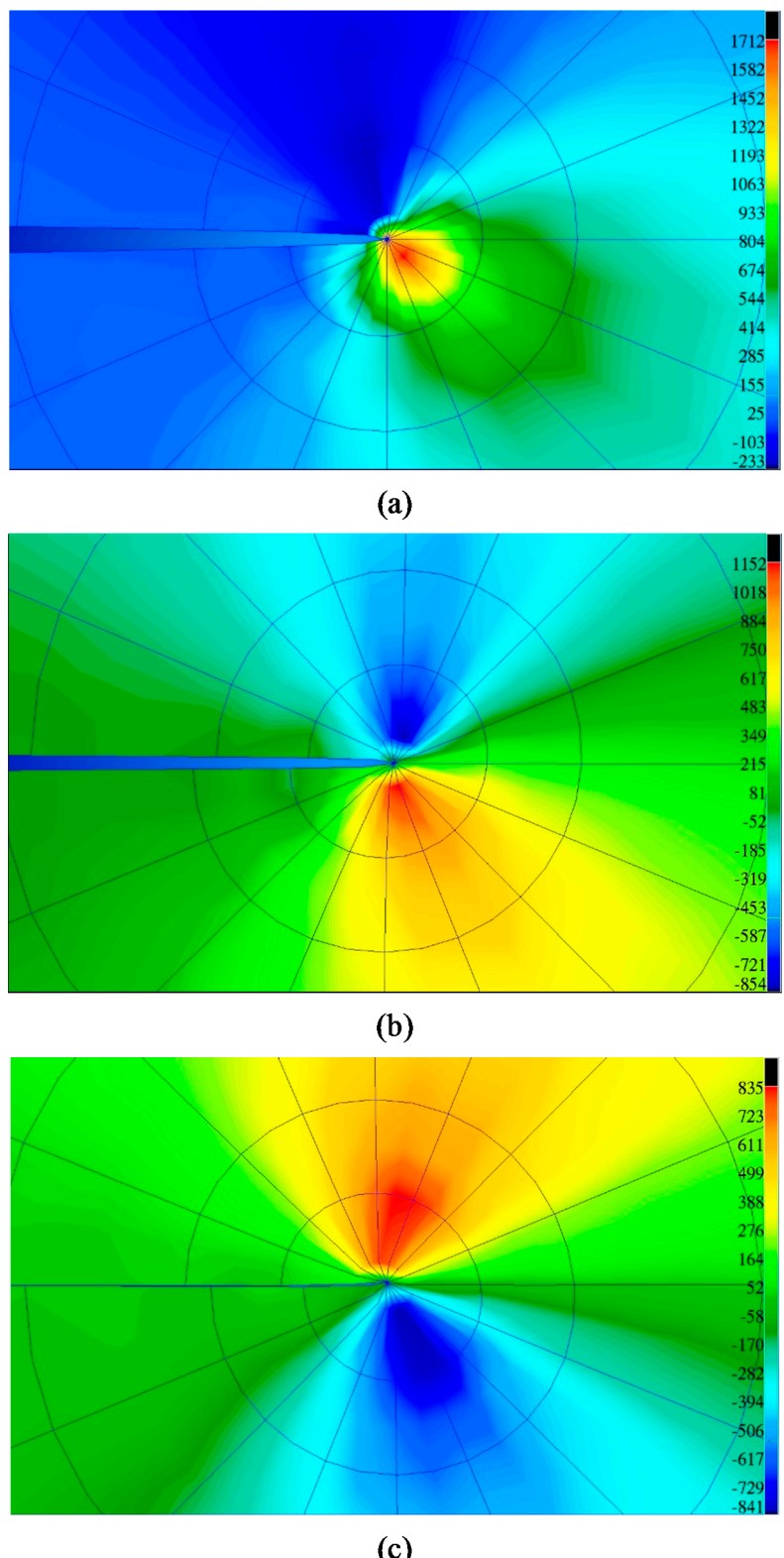

**Figure 14.** Tangential stress distribution near the crack tip for ARP2 ($\Delta K_{II}/\Delta K_I = 1.5$, $\delta = 120°$) at the (**a**) 90° (maximum $K_I$), (**b**) 150° (maximum $K_{II}$), and (**c**) 330° (minimum $K_{II}$); contour levels in MPa and the deformation magnification is 5×.

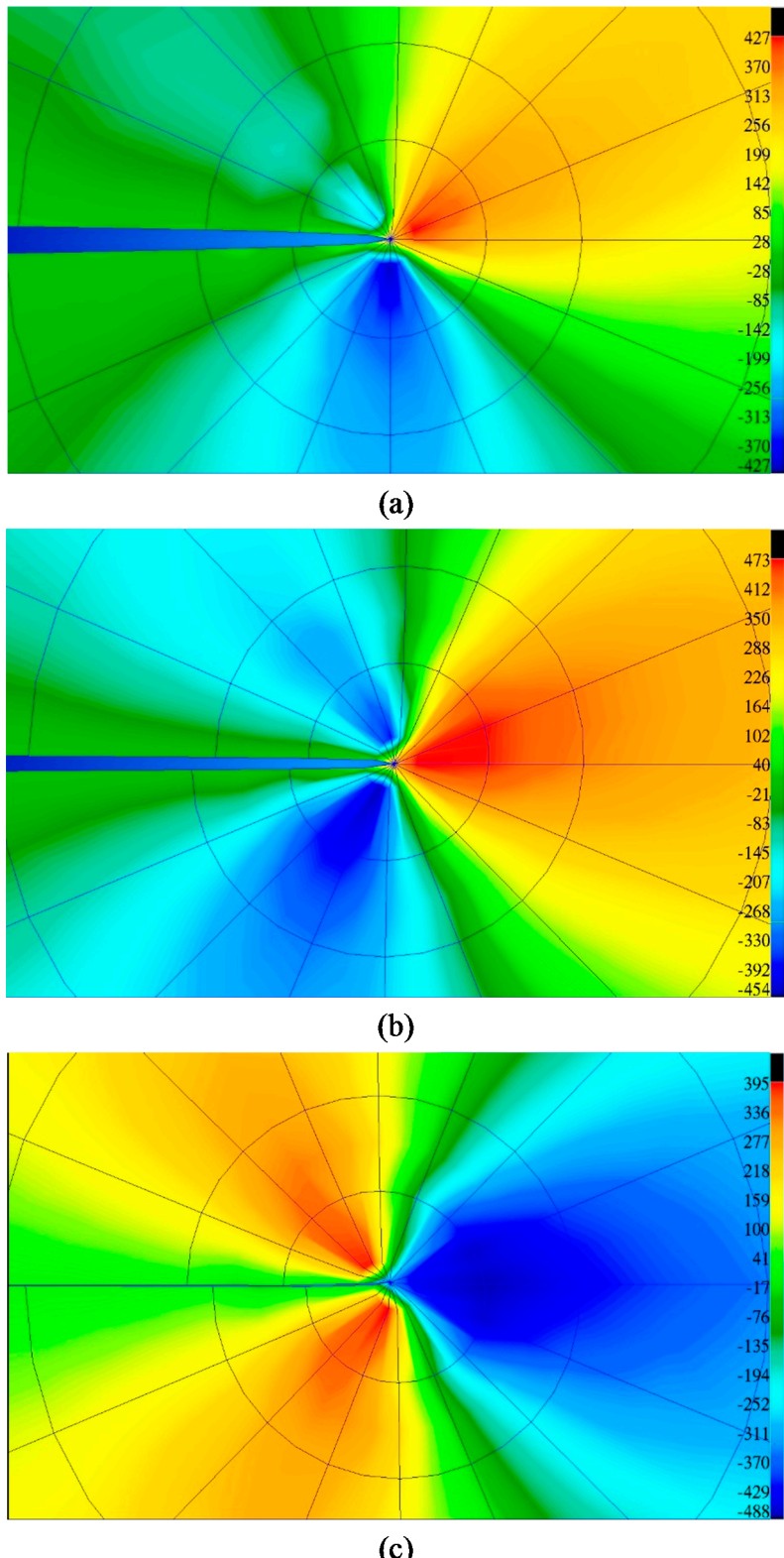

**Figure 15.** Shear stress distribution near the crack tip for ARP2 ($\Delta K_{II}/\Delta K_I = 1.5$, $\delta = 120°$) at the (**a**) 90° (maximum $K_I$), (**b**) 150° (maximum $K_{II}$), and (**c**) 330° (minimum $K_{II}$); contour levels in MPa and the deformation magnification is 5×.

### 3.2.1. Angles of Maximum Normal and Shear Stress Ranges

In this section, the purpose was to predict the incipient angle of crack growth after a change in the applied load from the condition, which leads the crack to stable coplanar growth. Dahlin and Olsson [13] suggested that the MTS range based on an elasto–plastic stress field could capture the effects of $\Delta K_{II}/\Delta K_I$ and $\delta$ on crack branching in rail steel. According to this criterion, the crack deviation angle lies along the material plane on which the tangential stress range ($\Delta \sigma_\theta$) is maximum during the selected load cycle. In this study, the $\Delta \sigma_\theta$ level on every plane was investigated; when $\sigma_\theta$ was below zero, it was reset to zero in the calculation. The maximum $\Delta \sigma_\theta$ value and the plane on which it appeared were indicated, respectively, as $\Delta \sigma_{max}$ and $\theta_{\sigma max}$. The shear stress range ($\Delta \tau_{r\theta}$) on every plane was also investigated. This underlying assumption was adopted: if the cracks grow via a shear mode, the growth direction should be determined by the plane ($\theta_{\tau max}$) on which $\Delta \tau_{r\theta}$ became maximum ($\Delta \tau_{max}$).

As mentioned in Section 2, $\Delta K_{II}/\Delta K_I$ and $\delta$ could influence the crack growth direction. More specifically, when $\Delta K_{II}/\Delta K_I$ increased from 1 to 1.5 with $\delta = 120°$, the crack growth changed from coplanar into branch; a $\delta$ increase from 90° to 120° while maintaining $\Delta K_{II}/\Delta K_I = 1.5$ changed the coplanar growth into branch growth; this was explicitly clarified by Dahlin and Olsson, namely, if this ratio was increased, the $\Delta \sigma_\theta$ direction switched from $\theta = 0°$ to $\theta = -70°$. Therefore, in this study, the $\delta$ influence on the crack growth direction was clarified.

The FEA results about the values of $\Delta \sigma_{max}/\Delta \tau_{max}$, $\theta_{\sigma max}$, and $\theta_{\tau max}$, when changing $\delta$ and maintaining $\Delta K_{II}/\Delta K_I = 1.5$, which is determined by comparing ARP1 and ARP2, are summarized in Table 6; when $\delta$ increased, $\Delta \sigma_{max}/\Delta \tau_{max}$ increased, whereas $\theta_{\sigma max}$ and $\theta_{\tau max}$ remained approximately on the branch and coplanar directions, respectively.

**Table 6.** Effect of $\delta$ on $\Delta \sigma_{max}/\Delta \tau_{max}$, $\theta_{\sigma max}$, and $\theta_{\tau max}$.

| No. | $\Delta K_{II}/\Delta K_I$ | $\delta$ (degree) | $\Delta \sigma_{max}/\Delta \tau_{max}$ | $\theta_{\sigma max}$ (degree) | $\theta_{\tau max}$ (degree) |
|---|---|---|---|---|---|
| ARP1 | 1.5 | 90 | 1.45 | −70 | 0 |
| ARP2 | 1.5 | 120 | 1.56 | −78 | 0 |

### 3.2.2. Crack Tip Opening and Sliding Displacements

The crack tip displacements are important since they could influence the crack growth rate and path direction. Therefore, the FEA results were used to obtain the crack tip opening displacement (CTOD), the crack tip sliding displacement (CTSD), and the residual crack tip opening displacement, which corresponded to the CTOD when the 50th loading was ended, of RP and RF. Figure 16 compares the resulting ranges of CTOD ($\Delta$CTOD), CTSD ($\Delta$CTSD), and residual CTOD of the two different materials (such as the comparison of ARP3 and ARF1). All these values were smaller for the RF case.

### 4. Discussion

The coplanar growth rates were plotted against $\Delta K_I$, $\Delta K_{II}$, and $\Delta K_{Ieff}$ for RP; the effective values were obtained considering that the crack closure and locking ratios were changed with respect to $\delta$. $\Delta K_{II}$ gave satisfactory correlations compared to $\Delta K_I$, suggesting that the crack growth rates were more influenced by mode II loading than by mode I loading. Moreover, $\Delta K_{Ieff}$ also provided relatively satisfactory correlations; however, these correlations were not sufficient because the crack growth contributions of mode I and II loading cycles were not mutually independent and, hence, the single mode range alone could not represent the crack growth rates uniquely. Therefore, the crack growth rates were plotted against $\Delta K_v$, which considered the interaction of the two loading cycles; although $\Delta K_v$ was invented for proportional loading, it gave satisfactory correlations.

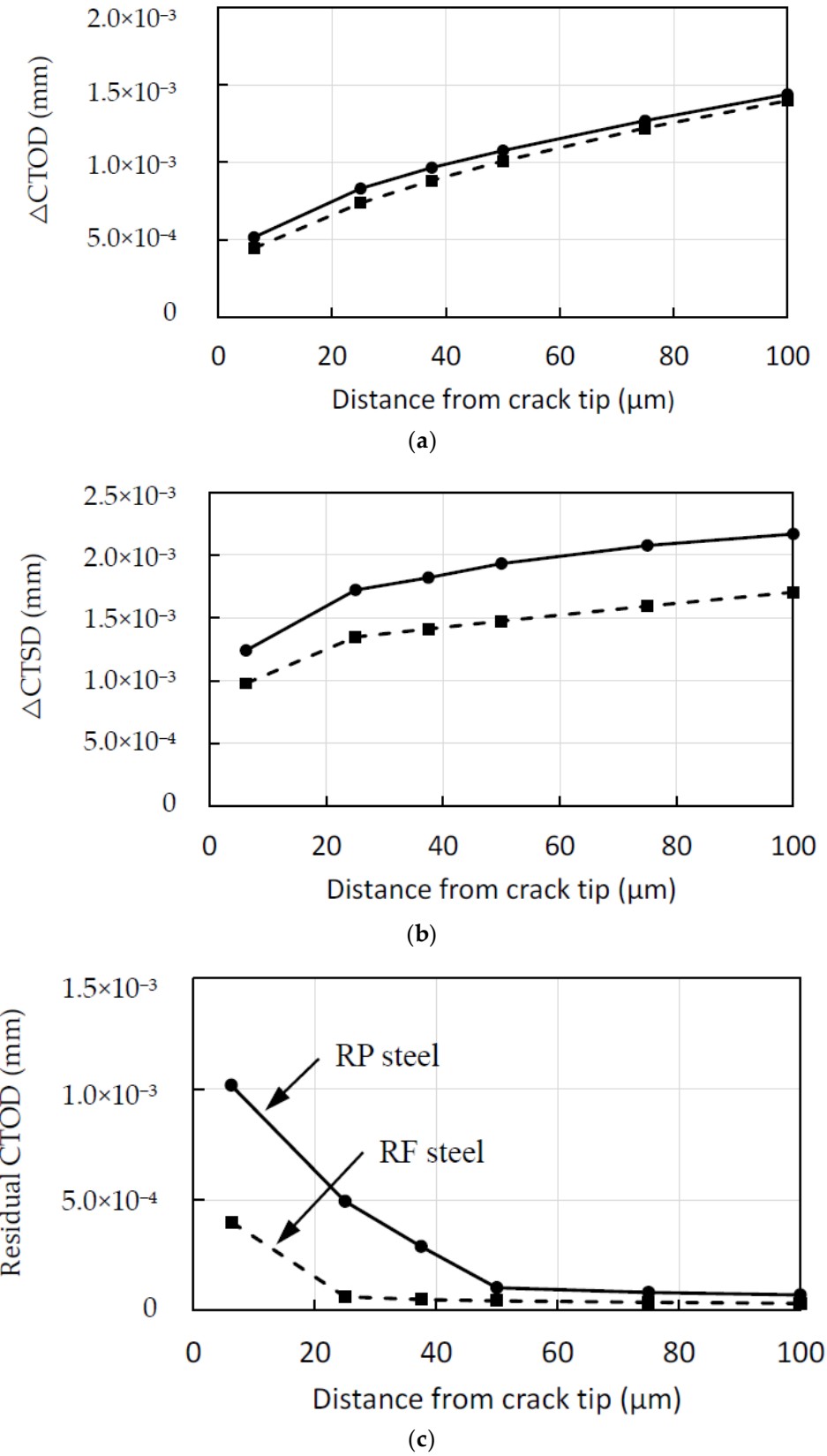

**Figure 16.** Ranges of (**a**) crack tip opening and (**b**) sliding and (**c**) residual crack tip opening displacement of RP (solid line) and RF (dashed line). ($\Delta K_{\mathrm{II}}/\Delta K_{\mathrm{I}} = 1.5$, $\delta = 30°$).

The branch crack growth rates could not be represented by a single line, as shown in Figure 11. All the cruciform specimens were tested under biaxial loading conditions and, in these cases, the non-singular stress called the *T*-stress acted at the crack tips. If the crack angle is not 45°, as in the branch crack case, the *T*-stress affects the plasticity near the crack tip and the plastic zone size increases with the shear part of the loading. Because a larger plastic zone yields higher growth rates [19], a separation of the data for $\Delta K_{II}/\Delta K_I = 2.5$ and $\Delta K_{II}/\Delta K_I = 2.0$ was observed, i.e., when the rate of shear loading was higher, the growth rates became higher.

The fractographic observation by SEM showed no clear striation patterns on the fracture surfaces near the crack tips, while roughness due to friction was observed in the case of RF1 and WT3. Such surface damage usually arises from the interaction between the crack faces under shear mode loading, which was reported in Fujii et al. [23]; therefore, the main crack growth mechanism observed in this study was assumed as caused by mode II loading.

The experimental results suggested that a $\delta$ increase could encourage crack branching. This can be elucidated by FEA; when $\delta$ increased, $\Delta\sigma_{max}/\Delta\tau_{max}$ increased and the $\Delta\sigma_{max}$ plane was oriented toward the branch direction, whereas the $\Delta\tau_{max}$ plane remained on the coplanar plane. The widely accepted criterion to predict the crack growth direction under non-proportional loading states that the crack selectively grows along the fastest growth direction. Based on this criterion, the FEA results can explain the experiment results.

The stresses were evaluated at the center of the elements around the crack tip, whose size was 25 μm. Because the MTS range criterion based on an elasto–plastic stress field is considered, these elements should be included inside the plastic zone ahead of the crack tip. The plastic-zone size under the investigated mixed mode I/II loading was not clear. Here, the cyclic crack tip plastic zone size ($r_p$) was derived from the analytical solutions based on pure mode I and mode II loading. Under the FEA conditions adopted, the size developed by mode II loading was larger than that resulting from mode I. If the size extension due to the stress redistribution is not taken into account, the size developed by mode II loading can be roughly estimated for both plane stress and strain conditions as

$$r_p = \left(\frac{1}{2\pi}\right)\left(\frac{\Delta K_{II}}{2\tau_{ys}}\right)^2 \tag{16}$$

where $\tau_{ys}$ is the yield stress of the material under shear. Based on all the conditions analyzed using RP, which were $\Delta K_{II} = 16.6$ MPa√m and $\tau_{ys} = 293$ MPa, the $r_p$ was calculated as 128 μm. Therefore, the points at which the stresses were evaluated were still far inside the plastic zone.

The stress evaluation was performed in the 50th loading cycle, although the steady-state stress distribution was not achieved because the distortion of elements increased, deteriorating the FEA accuracy. MARC provides a function to subdivide the mesh when the element distortion is large; one of the criteria is the maximum change in an interior angle from the initial angle for triangle elements and the recommended angle is 40°. For the ARP2 case, this criterion was violated at the 100th cycle and, hence, the stresses were evaluated at the 50th cycle with a margin.

However, remeshing was not performed in this study; the crack growth was expected to occur during the loading periods under the present testing conditions. In the ARP2 case, for example, $\Delta K_I$ and $\Delta K_{II}$ were 11.1 MPa√m and 16.6 MPa√m, respectively. Based on the crack growth law expressed in Equations (10) and (11), the crack growth length in 50 loading cycles was 3.4 μm, and the stress redistribution near the crack tip occurred according to this crack extension.

ΔCTOD and ΔCTSD were obtained for both RP and RF via FEA since they can influence the crack growth rate. In fact, a vector crack tip displacement (CTD) criterion has been proposed by Li [24] to correlate mixed mode FCG rates. This criterion was based on the concept that the vector CTD range (ΔCTD) is the driving force for FCG. Here, the vector ΔCTD corresponded to the vector summation of ΔCTOD and ΔCTSD. The FEA results indicated that ΔCTOD and ΔCTSD of RF were smaller than those of RP, probably because the former was a high-strength steel (Table 2); besides, RF exhibited much smaller residual CTOD and its opening distance under no loading compared to RP. Since there

were irregularities on the crack faces, there is a possibility that the mode II loading might have been attenuated, especially in RF. This implies that the crack growth rate in RF was lower than that in RP, explaining the experimental results.

## 5. Conclusions

Fatigue tests were conducted on rail and wheel steel specimens to obtain FCG rates under mixed mode I/II loading cycles, which can simulate rolling contact conditions, by using an in-plane biaxial fatigue machine. Simplified cycles were applied to the analysis of RCF cracks, including the effect of fluid trapped inside the cracks. To elucidate the experimental results, a FEA was also performed. The main findings can be summarized as follows.

(1) When the stress intensity range ratio between mode II and I increased, the crack tended to branch. This happened because, when this ratio increased, the direction of the maximum tangential stress range changed from coplanar growth to branch one.

(2) As the degree of overlap between the mode I and II stress intensities increased, the crack easily branched; in particular, the MTS range became superior to the maximum shear stress range and its plane oriented toward the branch direction.

(3) Coplanar crack growth rates were lower in head hardened rail steel than in normal rail steel. This was due to the smaller CTDs in head hardened rail steel, which can influence the growth rate, compared to normal rail steel.

(4) The branch crack growth rate varied considerably depending on the stress intensity range ratio even within the same rail steel and, unlike the coplanar growth rate, could not be correlated by a single line because the plastic zone size increased with the shear part of the loading as a result of the *T*-stress and this larger plastic zone gave higher growth rates.

(5) Clear striation patterns were not found near the crack tip region. Therefore, the main driving force for the crack growth under the investigated non-proportional mixed mode I/II loading was thought to be the mode II loading.

**Funding:** This research received no external funding.

**Acknowledgments:** The author thank Ippei Susuki of the Japan Aerospace Exploration Agency (JAXA) for his support in developing the test machine.

**Conflicts of Interest:** The author declares no conflict of interest.

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
