# Peer review of "Fatigue Crack Growth under Non-Proportional Mixed Mode Loading in Rail and Wheel Steel Part 1: Sequential Mode I and Mode II Loading"

_applsci, doi:10.3390/app9102006_

Round 1

Reviewer 1 Report

The manuscript investigates fatigue crack growth under Mode I/II for rail/wheel steels. The over all presentation and research approach are adequate, and the paper merits publication in Applied Sciences Journal after addressing the following comments: 1) Line 78: "previous studies on coplanar crack growth under non-proportional mixed mode I/II loading cycles apparently focused on rail steel" -Add a figure to show how Mode I/II present in both rail and wheel. A schematic figure should be sufficient. 2) Line 80: "Only Akama and Susuki [10] considered 78 also the wheel steel case, but their elucidation of the phenomena associated with crack growth under 79 such loading cycles was not sufficient" -Provide more details about the limitation of their study and highlight how this work bridge this gap in knowledge. 3) Add a paragraph before chapter 2 to describe the paper organization. 4) show and highlight the notch either in figure 1 or 2 5) Line 155, W is not shown in Figure-1 6) Line 282: change the text to "when analyzing fatigue problems" 7) Figure 11: change the notations in pictures from a,b to I,II. Also describe I,II by before and after crack propagation in the figure title.

Author Response

Response to Reviewer 1 Comments

Point 1: Line 78: "previous studies on coplanar crack growth under non-proportional mixed mode I/II loading cycles apparently focused on rail steel" -Add a figure to show how Mode I/II present in both rail and wheel. A schematic figure should be sufficient.

Response 1: A figure to show how Mode I/II present in both rail and wheel was add and the sentences were also added in the Introduction.

“Figure 1 shows how mode I and II loadings generated in the rail and wheel when the fluid penetrates into the crack. Details will be described in the next section.”

Point 2: Line 80: "Only Akama and Susuki [10] considered 78 also the wheel steel case, but their elucidation of the phenomena associated with crack growth under 79 such loading cycles was not sufficient" -Provide more details about the limitation of their study and highlight how this work bridge this gap in knowledge.

Response 2: The sentences as follows were added in the Introduction.

In particular, the reason why the growth direction of the crack changes depending on the loading conditions and the type of steel was not clarified…. The reliable criteria for predicting the crack path direction of non-proportional mixed mode loading were used, and the results were compared with corresponding experimental results.”

Point 3: Add a paragraph before chapter 2 to describe the paper organization.

Response 3: The sentences as follows were added before “2. Experiments”.

“This paper, part 1 of two companion papers, presents fatigue crack growth under non-proportional mixed mode I/II loading and is organized as follows. The introduction starts in this section with a brief overview of past papers for the crack growth under non-proportional mixed mode I/II loading cycles. Section 2 describes the detailed method of the experiments conducted and presents the experimental results. In section 3, the FEA model for predicting the crack path direction is presented and the results are indicated. Section 4 gives detailed considerations and discussions by comparing the experimental and FEA results. Finally, the important results obtained in this study are summarized in section 5.”

Point 4: show and highlight the notch either in figure 1 or 2.

Response 4: The notch was shown in Figure 3 (previous Figure 2).

Point 5: Line 155, W is not shown in Figure-1.

Response 5: I am sorry that “a plate of diagonal length 2W (refer to Fig.1)” was a mistake. Correctly, it was “a plate of diagonal length 2W (refer to Fig.3)”, and indicated “2W” in Fig.3.

Point 6: Line 282: change the text to "when analyzing fatigue problems"

Response 6: I changed the text to "when analyzing the fatigue problems".

Point 7: Figure 11: change the notations in pictures from a,b to I,II. Also describe I,II by before and after crack propagation in the figure title.

Response 7: I did.

Reviewer 2 Report

In this paper, fatigue crack growth rates of coplanar cracks and branched cracks are studied by combining experimental studies with finite element analysis. The research problem is clearly defined and the methods employed to address the research question are adequate. This paper can be accepted for publication after the following minor concerns are addressed.

1)     The authors should proofread the entire manuscript to improve the grammar.

2)     Conclusions 1 and 5 can be improved.

3)     From the fractographs, it is clear that high cycle fatigue is not found to be the ultimate fracture mechanism. The microscopic cup and cones indicate that the fracture is due to ductile fracture. The ductile fracture occurs when there are large stress concentrations along with large plastic strains. More information can be found in Kiran and Khandelwal (2014) “A micromechanics based cyclic void growth model”. Hence, the analysis presented in this study although is applicable to investigate crack growth and branching cannot be extended to predict ultimate fracture. It is important that the authors make a note of this fact in their paper to improve the scientific soundness of their work.

Author Response

Response to Reviewer 2 Comments

 Point 1: The authors should proofread the entire manuscript to improve the grammar.

Response 1: I am using an English proofreading service "Enago". If the reviewer thinks there is something wrong in English, please point out the specific portions.

Point 2:  Conclusions 1 and 5 can be improved.

Response 2: Conclusions 1 and 5 were modified as follows.

This happened because, when this ratio increased, the direction of the maximum tangential stress range changed from coplanar growth direction into branch one.”

Therefore, the main driving force for the crack growth under the investigated non-proportional mixed mode I/II loading was thought to be the mode II loading.”

Point 3: From the fractographs, it is clear that high cycle fatigue is not found to be the ultimate fracture mechanism. The microscopic cup and cones indicate that the fracture is due to ductile fracture. The ductile fracture occurs when there are large stress concentrations along with large plastic strains. More information can be found in Kiran and Khandelwal (2014) “A micromechanics based cyclic void growth model”. Hence, the analysis presented in this study although is applicable to investigate crack growth and branching cannot be extended to predict ultimate fracture. It is important that the authors make a note of this fact in their paper to improve the scientific soundness of their work.

Response 3: Thank you for introducing an excellent paper. I am very interested in micromechanics, and I have been conducting researches on microcrack initiation and growth. I would like to refer the paper you have introduced. For your suggestion, the following sentences in the text should be sufficient.

“3. Finite element analysis

3.1. Procedure

The elasto-plastic analysis was performed using the commercial FEA code MARC to better elucidate the crack growth characteristics. A stationary crack was considered because, in this study, a branch angle θ from the main crack was predicted.”

Reviewer 3 Report

The paper presents the results of an experimental campaign related to crack growth data from non-proportional mode I/II loading cycles. Empirical formulation is provided and a finite element analysis (FEA) was made. Although the paper is of interest and within the scope of the journal, the author should still better point out the innovation of this work and describe with more detail the FEA model.

Globally, the paper is well written and the results are well discussed. Nevertheless, there are some explanations that could be given along the text to allow the reader to better understand the results. The following comments should be taken into account before being considered for publication:

General comments:

1 – The title should be revised. There is no indication why this paper is “Part 1” and it only makes sense if a “Part 2” is published or being considered for publication.

2 – The number of specimens is arbitrary and may be insufficient to consider the variability of the material / phenomena. The readers should be made aware of that.

3 – Several explanations are given only in Section 4 (Discussion). This section is interesting and well structured, but the previous ones (especially those related to the presentation of results) are left with very low discussion and the readers are not aware of why a given choice is taken until Section 4.

Specific comments:

4 – Line 110: What were the limits for the pre-crack? Was this crack measured?

5 – Line 120 (Figure 2, a): How was the shape of the specimen obtained? Either provide references or indication of the shape study.

6 – Line 132 (Table 2): What does the author mean with Proof test? Is it tensile strength? Were the specimens for this proof tests obtained from the same batches of material that of those tested in the fatigue tests?

7 – Line 171: Explain why 4, 6 and 1 specimens were used for the tests? Is this number sufficient?

8 – Line 260: Why the consideration of two correlation lines? What is the difference of those 3 specimens? This information is only provided in lines 374 to 380 and that makes hard to the reader to understand Figure 10, when it is first seen.

9 – Line 298 to 301: How can the stresses distributions be validated? What was the benchmark for the FEA? How was the FEA model calibrated?

10 – Line 305: What does the author mean with “an accurate analysis”? What was the benchmark for this assumption?

11 – Line 346 to 349: What was the benchmark for the FEA model that allowed to use it in a parametric analysis?

12 – Line 354: The discussion of results on this section is insufficient and more information should be provided.

Author Response

Response to Reviewer 3 Comments

Point 1: The title should be revised. There is no indication why this paper is “Part 1” and it only makes sense if a “Part 2” is published or being considered for publication.

Response 1: I will soon submit a paper about sequential mixed mode I and mode III loading as part 2.  For the sake of clarity, I have declared this in the introduction as follows.

“This paper, part 1 of two companion papers, presents fatigue crack growth under non-proportional mixed mode I/II loading and …”

Point 2: The number of specimens is arbitrary and may be insufficient to consider the variability of the material / phenomena. The readers should be made aware of that.

Response 2: Although the number of specimens used may not be sufficient, experiments were conducted under different conditions for each specimen, and in all, a lot of conditions were performed.

Point 3: Several explanations are given only in Section 4 (Discussion). This section is interesting and well structured, but the previous ones (especially those related to the presentation of results) are left with very low discussion and the readers are not aware of why a given choice is taken until Section 4.

Response 3: I wrote to avoid repeating information within this paper, especially between results and discussions.

Point 4: Line 110: What were the limits for the pre-crack? Was this crack measured?

Response 4: I will rewrite the sentences “The starter notch was initially precracked by using an equibiaxial mode I loading with a zero stress ratio.” in more detail so that they are easy to understand as follows.

 “The starter notch was pre-cracked by using an equibiaxial mode I loading with a zero stress ratio. The load was reduced during pre-cracking in 10% steps until a fatigue crack growth rate of less than 10-9 m/cycle was reached. This was to reduce the residual plastic zone size to less than that produced by the first cycle of the test.”

Point 5: Line 120 (Figure 2, a): How was the shape of the specimen obtained? Either provide references or indication of the shape study.

Response 5: I will refer to Bold's doctoral dissertation as a reference for the specimen design.

Cruciform specimens (Fig.3) having the 45° starter notch with a half-length of 2 mm, made by spark erosion, were used in the experiments [15]. …

15. Bold, P. E. Multiaxial Fatigue crack growth in rail steel. PhD Thesis, University of Sheffield, Sheffield, UK, 1990.

Point 6: Line 132 (Table 2): What does the author mean with Proof test? Is it tensile strength? Were the specimens for this proof tests obtained from the same batches of material that of those tested in the fatigue tests?

Response 6: For some ductile metals, yield stress is difficult to define and calculate, so proof stress is made. The values in Table 2 are all catalogue values. I will replace “Proof stress” in Table 2 and corresponding main text with “0.2% proof stress” to avoid misunderstanding.

Point 7: Line 171: Explain why 4, 6 and 1 specimens were used for the tests? Is this number sufficient.

Response 7: Please see in Response 2.

Point 8: Line 260: Why the consideration of two correlation lines? What is the difference of those 3 specimens? This information is only provided in lines 374 to 380 and that makes hard to the reader to understand Figure 10, when it is first seen.

Response 8: I will add the following sentence to the part concerned.

“As can be seen, the crack growth rate differs considerably depending on ΔKII/ΔKI, and increasing in the ΔKII/ΔKI increases the crack growth rate.”

Point 9: Line 298 to 301: How can the stresses distributions be validated? What was the benchmark for the FEA? How was the FEA model calibrated?

Response 9: In “3. Finite element analysis  3.1. Procedure”, there is a following sentence: “The use of this mesh and loading procedure allowed the confirmation that the MTS occurred at θ = –70.5°due to pure mode II loading in the elastic analysis.”

No model calibration was performed. Generally in FEA, it is difficult to obtain an accurate absolute values. Instead in this paper, various physical quantities were compared by relative values.

Point 10: Line 305: What does the author mean with “an accurate analysis”? What was the benchmark for this assumption?

Response 10: As often said, the serendipity element in its angular-distorted configuration cannot represent quadratic displacement fields exactly (please refer to, for example, “Effects of element distortions on the performance of isoparametric elements, Nam-Sua Lee and Klaus-Jurgen Bathe”, Int. J. Num. Meth. Eng., Vol. 36, pp.3553-3576 (1993)). Based on these results, MARC provides a function to subdivide the mesh when the element distortion is large; one of the criteria is the maximum change in an interior angle from the initial angle for triangle elements and the recommended angle is 40°. No numerical study was conducted based on the benchmark problem.

Point 11: Line 346 to 349: What was the benchmark for the FEA model that allowed to use it in a parametric analysis?

Response 11: Strictly speaking, as reviewer pointed out, FE mesh calibration should be performed based on the benchmark problem, however, no numerical research was conducted in this study.

Point 12: Line 354: The discussion of results on this section is insufficient and more information should be provided.

Response 12: As I responded previously, I wrote to avoid repeating information within this paper, especially between results and discussions. The discussion of the results was done only in Section 4.

Round 2

Reviewer 3 Report

The reviewer thanks the authors for respecting the reviewer's comments and by implementing the asked information in the updated version of the manuscript. The present version has been significantly improved.

Author Response

Thank you very much for your detailed review. My paper has been significantly improved by your sincere review.